# Therapeutic Potential in Wound Healing of Allogeneic Use of Equine Umbilical Cord Mesenchymal Stem Cells

**DOI:** 10.3390/ijms25042350

**Published:** 2024-02-16

**Authors:** Ailén Iribarne, María Belén Palma, Laura Andrini, Fernando Riccillo, Delfina Rodriguez, Martín Casella, Felipe Garay, Julieta Spoto Zabala, Leandro Mazza, Adriana Muro, Guillermo Buero, Santiago G. Miriuka, Edgardo Carosella, Marcela N. García

**Affiliations:** 1Cátedra de Citología, Histología y Embriología, Facultad de Ciencias Médicas, Universidad Nacional de La Plata, Buenos Aires CP 1900, Argentina; airibarne@med.unlp.edu.ar (A.I.); mbpalma@med.unlp.edu.ar (M.B.P.); landrini@med.unlp.edu.ar (L.A.); friccillo@fcnym.unlp.edu.ar (F.R.); 2Instituto de Neurociencias (INEU), Fundación para la Lucha contra Enfermedades Neurológicas de la Infancia-Consejo Nacional de Investigaciones Científicas y Técnicas (FLENI-CONICET) Escobar, Buenos Aires CP B1625, Argentina; smiriuka@fleni.org.ar; 3Centro de Reproducción Equina La Adela, Chascomús, Buenos Aires CP 7130, Argentina; 4División Remonta Haras La Teruca, Buenos Aires B1900, Argentina; 5Dirección Veterinaria, Policía de la Provincia de Buenos Aires, Buenos Aires B1904, Argentina; 6Cátedra de Anatomía B, Facultad de Ciencias Médicas, Universidad Nacional de La Plata, Buenos Aires CP 1900, Argentina; drleandromazza@gmail.com; 7Laboratorio de Análisis Clínicos Veterinarios, La Plata, Buenos Aires CP 1900, Argentina; 8Sanatorio Mater Dei, Ciudad Autónoma de Buenos Aires, Buenos Aires CP C1425, Argentina; wbuero@gmail.com; 9Commissariat a l’Energie Atomique et aux Energies Alternatives (CEA), Service de Recherche en Hemato-Immunologie (SRHI), Saint-Louis Hospital, 75010 Paris, France; edgardo.carosella@cea.fr; 10Hopital Saint-Louis, IUH, Université Paris Diderot, Sorbonne Paris Cite, 75010 Paris, France

**Keywords:** horses, wound healing, mesenchymal stem cells, cellular therapy, equine umbilical cord

## Abstract

Wound healing after skin injury is a complex process, particularly in equines where leg wounds are prevalent and their repair is complicated due to the anatomical characteristics. Conventional treatments are not effective enough. The umbilical cord offers an unlimited source of adult mesenchymal stem cells (ucMSCs) from Wharton’s jelly tissue. The present study aims to demonstrate the safety and therapeutic potential of the allogeneic use of equine ucMSCs (e-ucMSCs) in the healing of severe equine leg wounds. The methods employed were the isolation, culture and expansion of e-ucMSCs. Flow cytometry and a PCR assay were used for cell characterization. This study included an immunomodulation assay, a murine pre-clinical trial and the first phase of an equine clinical trial. Our results showed that e-ucMSCs express a functional HLA-G homolog, EQMHCB2. In the immunomodulation assay, the e-ucMSCs inhibited the proliferation of activated equine peripheral blood mononuclear cells (e-PBMCs). In the murine pre-clinical trial, e-ucMSCs reduced healing time by 50%. In the equine clinical trial, the injection of e-ucMSCs into severe leg lesions improved the closure time and quality of the tissues involved, regenerating them without fibrous tissue scar formation. In conclusion, the results of this study suggest that e-ucMSCs can be used allogeneically for wound healing by creating a tolerogenic environment.

## 1. Introduction

Wound healing after skin injury involves a complex process that affects different cellular components. In normal physiological conditions, the restoration of functional tissue occurs. Nevertheless, some conditions can interfere with this process, leading to slow healing and eventually to the appearance of extended fibrous scars. Different pathologies are often observed in horses, especially leg injuries characterized by delayed re-epithelialization and a fibroproliferative response [1]. Equine athletes frequently have severe skin lesions on the lower limbs that can become infected due to the lack of complete and rapid healing. Even more complicated are wounds involving other tissues, such as muscles and tendons. These injuries can result in high morbidity and often compromise full recovery [2]. Usually, after a severe injury, the repair mechanism of the skin and surrounding tissues is based on a fibrotic process, in which the scar tissue formed is functionally deficient and tends to become stiffer, with significant consequences for the animal in terms of reduced performance [3]. In this type of injury, healing is a slow process that, when complete, generates fibrotic tissue with an increased risk of re-injury, also affecting the daily life of the horse. 

Mesenchymal stem cells (MSCs) have great potential for regenerative medicine given their potent immunomodulatory and pro-regenerative properties [4]. Previous reports confirmed that MSCs secrete soluble factors that suppress peripheral blood mononuclear cell (PBMC) proliferation and alter cytokine expression to modulate the immune response [5]. Therefore, the immunomodulation generated by MSCs can decrease the local inflammation process and help to prevent transplant rejection. In addition, these cells can upregulate endogenous cells, reactivating the injured microenvironment. Both properties may explain the long-lasting regenerative effects of MSCs, which are useful for long-term treatment [6].

MSCs can be harvested from different sources including bone marrow and adipose tissue followed by ex vivo expansion. Their potential clinical applications in multiple disorders, such as orthopedic injuries and inflammatory diseases, are being investigated [7,8]. Current results with MSCs are promising. It has been published that, especially in the case of equine superficial digital flexor tendon injuries, the risk of recurrent injuries is reduced by 55–25% [3]. The latest clinical practices use autologous MSCs from bone marrow or adipose tissue. However, this approach requires the aspiration of bone marrow from each horse, an invasive technique with a risk of potential complications, such as systemic infections. Another factor to consider is that bone marrow MSC number decreases in older animals [5,9]. On the other hand, MSCs are also obtained from adipose tissue by needle biopsy or by liposuction aspiration, which can be risky and difficult due to the small amount of accessible fat in highly trained athletic horses [9,10].

Many of these limitations of current autologous treatment could be overcome by the use of allogeneic MSCs, as they offer the advantage of near-immediate cell therapy for the treatment of acute orthopedic lesions without the period associated with the isolation and expansion of autologous MSCs [9]. Previous trials have shown that allogeneic equine bone marrow MSCs can be transplanted into the injured tendon, injected intradermally or intraarticularly without eliciting an apparent immune response. Additionally, no changes in cellular or humoral immunity parameters were reported after intravenous injection of allogeneic MSCs into healthy horses [5,9,11].

Unlike the other sources of MSCs, the umbilical cord provides an unlimited source of adult MSCs (ucMSCs) from Wharton’s jelly tissue with identical attributes relevant to clinical applications. ucMSCs are reported to be highly proliferative and have potent immunomodulatory properties. Furthermore, the process of obtaining them does not require any practice that causes pain or some traumatic consequences, as the umbilical cord is routinely discarded upon delivery [12,13]. 

Previously published articles have reported that in humans, ucMSCs express the Human leukocyte antigen G (HLA-G), a non-classical MHC-I molecule, which plays an important role in maternal tolerance of the fetus [14]. Horses differ from humans in that the fetal chorion does not come into contact with circulating maternal blood (epitheliochorial placenta). However, immunological communication involving MHC-I also appears to play a role in pregnancy and parturition [15]. Equine ucMSCs (e-ucMSCs) express a protein, called EQMHCB2, which could be a functional ortholog of HLA-G, and its mRNA is upregulated in the allantochorion during labor [15]. High levels of expression of the equine HLA-G ortholog protein in ucMSCs could be seen as an advantage in the treatment of wounds, since, as we have shown in previous experiments with human ucMSCs, they would modulate the local activity of the immune system, helping to prevent rejection of the cell transplant [12].

A deeper understanding of equine ucMSC biology is critical to guide veterinary clinical trials. In addition, a successful treatment in horses should be considered for its possible extrapolation to clinical trials in humans with similar pathologies [6,7,8]. Given the crucial importance of developing a novel therapy capable of restoring functional tissue in severe wounds in horses, the main purpose of this study is to establish the safety and therapeutic potential of the allogeneic application of e-ucMSCs in the field of wound healing.

## 2. Results

### 2.1. E-ucMSC Characterization

By flow cytometry analysis, we checked cell surface molecules that specifically characterize human MSCs. The e-ucMSCs were analyzed in passage 1 for CD29, CD73, CD90 and CD98; however, equine ucMSCs were only positive for CD90 (Figure 1). This analysis is important as it demonstrates the persistent undifferentiated state of the cells once isolated from the umbilical cord microenvironment. 

### 2.2. Inhibitory Effect of e-ucMSCs on e-PBMC Proliferation

E-ucMSCs were co-cultured with activated equine peripheral blood mononuclear cells (e-PBMCs) to assess whether these cells have any regulatory effect on immune cells. In Figure 2, we can observe that the mean proliferation percentage of activated e-PBMCs was significantly lower when they were co-cultured with e-ucMSCs. When the e-PBMCs were stimulated and cultured alone, the number of cell divisions reached five and the percentage of non-proliferating cells was 46%, whereas when the e-PBMCs were exposed to e-ucMSCs, the number of cell divisions diminished and the percentage of non-proliferating e-PBMCs increased to 61%. In addition, Figure 3 shows that at each cell division, the percentage of proliferating e-PBMCs was always significantly lower in the presence of e-ucMSCs than in their absence. 

### 2.3. Expression of Equine HLA-G Homologs in e-ucMSCs

To explain the immunosuppressive effect of e-ucMSCs observed above, we decided to evaluate if they express the ortholog of *HLA-G*, called *EQMHCB2*.

From a single pre-mRNA, alternative splicing can form seven distinct HLA-G isoforms, of which four are membrane-bound isoforms (HLA-G1 to -G4) and three are soluble (-G5 to -G7). The mRNA obtained from e-ucMSCs was analyzed by RT-PCR using different primer sets to evaluate the expression of proteins similar to some of the HLA-G isoforms (shown in Table 1). The results showed that, for primer set C, e-ucMSCs presented a band of 120 bp in 60% of the cases, while for the positive control (human cell line, JEG-3), the band observed was 150 bp. By aligning the sequence of the horse gene (*EQMHCB2*) with the human one, it could be observed that the first has several gaps, which would explain why the band in e-ucMSCs has fewer bp (see the green consensus identity line shown in Figure 4). Considering the number of bp, the equine isoform found could be similar to the human HLA-G3 isoform, which contains only α1 domain (Figure 4).

### 2.4. Xenograft Transplantation of e-ucMSCs: Pre-Clinical Assay in Mouse Skin Model

The use of laboratory animal tests offers an excellent option to evaluate e-ucMSCs’ advantages and disadvantages. The tolerance of e-ucMSCs was investigated in vivo by xenotransplantation of the cells into immunocompetent C3H/s mice. For this purpose, skin wounds were made on the flank of mice, affecting the epidermis, dermis and subcutaneous tissues. Two groups of mice were studied: a control group (injected only with PBS) and a treated group (injected with e-ucMSCs). The evolution of the dermal lesion was monitored over 25 days. Comparing the control and treated groups, we observed that e-ucMSCs improved wound healing and hair recovery (Figure 5).

In the control group, the normal wound healing process took an average of 21 ± 1.5 days. In contrast, mice treated with a single dose of e-ucMSCs showed significantly accelerated healing, achieving complete healing in an average of 11 ± 1 days. Thus, the treatment allows for a remarkable 50% reduction in wound closure time. These results agreed with the hair recovery, which took 25 ± 1.13 days in the control group and 13 ± 1 days in the group treated with e-ucMSCs (Figure 6). The observed differences in wound healing and hair recovery parameters were statistically significant between groups (*p* < 0.01). In both groups no signs of local inflammation were observed throughout the trial, and particularly, in the treated group, no external signs of cell rejection were observed.

In addition, a quantification of the percentage of the wound healing area at different time points after the e-ucMSC injection was performed. Figure 7 shows that there were significant differences in the three post-injection times analyzed between the control group and the treated group. The highest differences between groups were observed on the third day, when the mice that had received a dose of e-ucMSCs had already closed 75% of their initial wound area, while the control group only achieved 30% closure. At 10 days post-treatment, even though the differences between groups are minor, it was observed that the treated mice had made significantly more progress in the healing process, reaching wound healing rate values of close to 100%, clearly showing the improvement provided by the e-ucMSCs.

The possibility of applying several doses was raised for the treatment of larger wounds that therefore required a longer treatment. For this purpose, a trial was carried out to check the local reaction of e-ucMSCs with four serial doses, in which 250,000 e-ucMSCs/100 μL of PBS were injected each time around the wound. No local reactions such as swelling or redness lasting more than 24 h after each injection were observed. In addition, skin samples were taken after the last application, and no lymphocyte infiltration or tissue abnormality was observed. 

### 2.5. Allogeneic Transplantation: Clinical Trial on Injured Horses

In the equine clinical trial, four horses with self-inflicted wounds were treated at the Horse Ranch. These horses had severe and deep wounds on their legs. In case 1, the horse had a cut injury on the right forelimb that affected the synovial sheath. In the second case, the injury on the right forelimb had broken fascia causing the muscle to separate, exposing the bone (Figure 8). Cases 3 and 4 presented injuries to the hind limbs with muscle swelling in which the tendon was exposed and the common extensor digitorum sheath was injured. Taking into account the severity and extent of the injuries, in agreement with the veterinary staff, a second application of e-ucMSCs was performed on the first two cases. In all cases treated with e-ucMSCs, we observed a significant improvement in wound healing.

Blood tests performed after cell injection showed no significant changes in health parameters. To illustrate this, Table 2 shows the representative results obtained from case 2. Before cell treatment was started, white blood cells (WBCs) were higher than normal, and after the administration of one or two doses of cells, the values normalized. The decrease in WBCs to normal values after the administration of the allogeneic cells indicates that there was no cellular rejection by the host. Furthermore, it can be observed that the WBC relative percentage remains stable throughout the treatment. These results are consistent with the macroscopic observation, where no signs of local inflammation were found after e-ucMSC injection.

The observed improvement was not limited to the superficial layers of the skin, but a faster recovery of the other affected tissues was also evident. Wound healing time was significantly shorter than historical for this type of leg injury. Finally, when the wound closes, only a small scar remains on the skin, with no fibrosis. The surrounding tissue is completely regenerated, as evidenced by the appearance of new hair follicles and hair growth in Figure 8. 

To conclude these studies, once the lesions had closed, ultrasound scans were carried out to check the regeneration of the deeply involved tissues. For additional information, some examples of blood tests and ultrasound scans can be found as annexes (shown in Appendix A).

## 3. Discussion

Failure to restore a functional epidermal barrier can lead to the establishment of a fibrous skin defect. In horses, skin lesions of the limbs are very common, and their repair is complicated due to the anatomical features. Severe injuries involving epidermal and connective tissue, and even tendons and muscles, can become disabling for an animal, but even so, there are currently no treatments deemed satisfactory [16]. This failure means that these severe injuries often take months to heal and result in large fibrous scars that can affect the performance of a sporting equine. In this context, therapy with MSCs has shown to be a promising therapeutic strategy [7,15,17]. The use of autologous MSCs has been successful despite the major limitations mentioned above. In contrast, the ucMSCs have enormous therapeutic potential given their particular characteristics. Umbilical cords are obtained from placentas discarded in normal deliveries, thus avoiding any traumatic procedures and potential complications in the collection of cells. In addition, a large number of cells, expandable in culture, are obtained in this way [18]. The ucMSCs are multipotent cells and can differentiate into mesenchymal lineages, such as osteocytes, chondrocytes and adipocytes. This feature was studied by the multipotency assay and published in a previous article [19]. Considering this, the presence of ucMSCs in the wound could be effective not only for promoting cell proliferation from the adjacent tissues, but also for differentiating into these mesenchymal cell populations. 

It is important to highlight that our results concerning surface markers (CDs) showed that the e-ucMSCs were positive for CD 90 and negative for other markers of cellular undifferentiation. This could be because the commercially available antibodies used in flow cytometry are for human cells, so some interspecies differences in CD conformation would explain the difficulty in recognition. These results are in concordance with those of Cequier et al. who published that there exists a great possibility of variability in CD expression in equine ucMSCs [8].

As we published previously, in human ucMSCs, another relevant aspect is the high level of HLA-G expression [20,21]. In horses, the HLA-G homologous immune checkpoint has not yet been studied in detail. However, considering that most cellular mechanisms that are efficient are usually evolutionarily conserved, it could be likely that e-ucMSCs express a protein with structure and function similar to HLA-G. The cells were analyzed by RT-PCR with different primers used to detect HLA-G, and we were able to observe the expression of a similar protein present in equine cells, named EQMHCB2, which was published in a previous paper [15]. By aligning the sequence of the horse gene (*EQMHCB2*) with the human one, we consider that it is similar to the HLA-G3 isoform. The expression of this immune checkpoint is important for the acceptance of allogeneic cells due to its modulating effect on the local immune response. The results observed in the co-culture between e-ucMSCs and e-PBMCs confirm that e-ucMSCs have an important immunosuppressive effect by inhibiting the proliferation of activated lymphocytes. 

Stem cell therapy activates several key cellular processes for damaged tissue regeneration. These include the contribution of enabling parenchymal cells, whose action promotes angiogenesis and induces paracrine action on neighboring cell populations. All these factors contribute to the regeneration and repair of injured tissues [20,22,23]. As we have already mentioned, angiogenesis plays a key role in wound healing, being especially important in extended skin lesions, by restoring blood perfusion and delivering nutrients to injured sites [23]. Taking into account all the aforementioned properties, the e-ucMSC cell line emerges as an excellent candidate for wound healing treatment [24].

To evaluate the safety and efficiency of e-ucMSC-based therapies, we have used immunocompetent C3H/s mice in a representative pre-clinical in vivo model. Previous works have reported that mice have another functional and structural equivalent of HLA-G, called Qa2 [25]. We considered, therefore, that EQMHCB2 is also a functional homolog of HLA-G and could probably exert its action in murine lymphocytes. In the present study, the immunomodulatory e-ucMSC activity was further demonstrated by their effect as xenotransplants in immunocompetent C3H/s mice without any immunosuppressive treatment. Due to this functional homology, equine cells injected into mice can induce immunosuppressive effects, very similar to what happens with human ucMSCs [12], avoiding cellular rejection by the host. The results of the in vivo assay were highly positive since the e-ucMSC population significantly diminished the period required for wound healing. Although the experimental lesion produced in mice is not due to the common pathogenesis observed in horse’s severe skin defects, the healing and regeneration processes of the dermis and epidermis are similar, which made it possible to predict the future behavior of the e-ucMSC population for its application in equine lesions. Histological analysis of the skin murine sections confirmed this observed improvement in wound healing and the absence of an associated inflammatory process when e-ucMSCs were transplanted, as was also observed in a previous study after the application of human ucMSCs [12]. The repair and remodeling process after e-ucMSC application showed not only faster but also better regeneration patterns, for example in collagen fibers, than those observed in the control group. 

All these previous results allowed us to design the first clinical trial in equines. For this purpose, four horses with severe leg injuries were selected in agreement with the veterinary staff. The injuries had been caused by the fencing of the different sections of the Horse Ranch. This type of injury is unfortunately common in sporting or working equines due to their impetus and personality. Satisfactorily, in all cases, the e-ucMSC treatment worked as expected. Importantly, e-ucMSCs were not recognized as foreign or rejected when applied allogeneically in horses, and they also significantly improved wound repair. This improvement was not only in terms of time, which was significantly reduced, but also in the regeneration of the tissues involved, leading only to a much smaller final scar than expected with conventional treatments. Although several studies have been performed on equine cell therapy with e-MSCs [26,27], there are few reports that use umbilical cord MSCs. This article is the first to focus on the allogeneic application of e-ucMSCs in clinical cases for wound regeneration. 

Our preliminary trial aimed to understand the effect of these cells under different conditions; therefore, clinical cases with differences in the extent, depth and types of tissues involved were selected. For this reason, we consider this work as a baseline for more comprehensive clinical trials. Future studies could increase the number of cases analyzed and evaluate wounds with the same severity, using controls to allow a more adequate comparison of wound closure times and wound healing quality.

It is important to note that a successful treatment in horses should be considered for its possible extrapolation to clinical trials in humans with similar pathologies, taking into account that this species is one of the most similar in terms of its characteristics and types of injuries. Finally, no less important is the fact that these cells are obtained from disposable tissue, so their manipulation and therapeutic use do not bring any ethical objection.

We can conclude that the e-ucMSC population has undergone comprehensive characterization in both in vitro and in vivo assays that demonstrate its safety and efficacy for the proposed therapy. The data reported here indicate that the e-ucMSC population studied allows allogeneic transplantation and has positive effects on wound healing. Consequently, e-ucMSCs may assume a pivotal role in regenerative medicine, emerging as a novel cell-therapeutic tool, particularly for dermal healing disorders in equine limbs. This cell line could generate a promising new alternative therapy that reduces convalescence time and rapidly restores an equine to an athletic state, with a lower risk of re-injury.

## 4. Materials and Methods

### 4.1. Collection of Equine Umbilical Cords

Equine umbilical cords were obtained from natural deliveries. Samples were disinfected with 5% sodium hypochlorite solution, washed 3 times with physiological solution and placed in bottles containing a transport medium, Dulbecco’s phosphate-buffered saline (PBS, Sigma-Aldrich, St. Louis, MO, USA, #P3813) + penicillin/streptomycin/gentamicin. Finally, they were transported in cold containers to the Cell Culture Laboratory for cell isolation and expansion. 

### 4.2. Isolation and Expansion of e-ucMSCs

Samples were disinfected again in the laboratory with 5% sodium hypochlorite solution and washed 3 times with PBS. For cell isolation, each cord was chopped into 5 mm long fragments, a sagittal cut was performed to expose Wharton’s jelly and umbilical blood vessels were carefully removed with clamps. These fragments were washed 2 or 3 times with PBS to remove the remaining blood. The side with the exposed jelly was then placed against the bottom of the culture plate, and minimum essential medium (α-MEM, Gibco, New York, NY, USA, #A1049001) + 20% fetal bovine serum (FBS) + penicillin/streptomycin/gentamicin was added. The plates were incubated at 37 °C in a humid atmosphere and 5% CO_2_. The cell culture medium was changed every 2–3 days. After the first week, gentamicin was removed from the culture medium. Expansion of e-ucMSCs was observed 10–14 days after explantation, and these cells were amplified to 2/3 passage.

### 4.3. Flow Cytometry (FC) Analysis

The specific surface molecules of e-ucMSCs were characterized by flow cytometry analysis. Cells were dissociated by incubation with trypsin and conjugated with the following specific cell surface markers: CD73, CD29, CD90 and CD98. CD73-PE: BD Biosciences, Franklin Lakes, NJ, USA, # 55027; CD29-PE: BD Biosciences, Franklin Lakes, NJ, USA, #561795; CD90-FITC: BD Biosciences, Franklin Lakes, NJ, USA, #555595; and CD98-FITC: Molecular Probes, Eugene, OR, USA, #A15739, were used as markers. Incubation was carried out in PBS + 0.5% bovine serum albumin for 30 min at room temperature, with the primary antibodies (1:50 dilution). FC analysis was performed on a BD Accuri cytometer. Data were analyzed with FlowJo 7.6 software.

### 4.4. Immunomodulatory Capacity Assay of e-ucMSCs

The immunomodulatory activity of e-ucMSCs was demonstrated by the inhibition of lymphocyte proliferation ex vivo. Equine peripheral blood mononuclear cells (e-PBMCs) were freshly isolated from buffy coat leukocyte concentrates obtained from healthy horses. Peripheral blood was diluted 1:2 with PBS. The isolation of e-PBMCs was carried out by density gradient centrifugation using Ficoll-Paque PLUS medium (Fischer Scientific, Roskilde, Denmark, #17-1440-03). Subsequently, e-PBMCs were stained with CellTrace CFSE Cell Proliferation Kit (Invitrogen, Waltham, CA, USA, # C34554), a vital dye that loses fluorescence after each cellular division. The e-PBMCs were then incubated with 5 mg/mL concanavalin-A and 300 IU/mL interleukin-2 for activation and then co-cultured for 4 days with e-ucMSCs using Roswell Park Memorial Institute 1640 medium (Gibco, New York, NY, USA, #61870036) + 10% FBS. After incubation, lymphocytes were analyzed by FC, and cycle counting was performed.

### 4.5. RNA Extraction, cDNA Synthesis, and RT-PCR Analysis

To confirm the presence of an equine protein, a homolog of human HLA-G, reverse transcription polymerase chain reaction (RT-PCR) was performed using specific primers to detect all known human isoforms. The primers used were published previously [28]. RNA extraction was performed with TRIzol Reagent (Invitrogen, CA, USA, #15596026). RNA purity and concentration were assessed with a Nanodrop spectrophotometer. For cDNA synthesis, 500 to 1000 ng of total RNA was retrotranscribed with MMLV reverse transcriptase (Promega, Madison, WI, USA, #M170A), according to the manufacturer’s instructions. PCR was performed with Dream Taq Green PCR Master Mix 2X (Thermo Scientific, Waltham, MA, USA, #K1081). The oligonucleotides are published in Table 1.

### 4.6. Animals

Immunocompetent adult mice, C3H/S strain, inbred in the Biotherium of the Histology Chair of the School of Medical Sciences were used. The mice were placed in individual boxes, with an alternating light cycle of 12 h of illumination and 12 h of darkness (Circadian Rhythms Room), with forced ventilation and a temperature of 22 ± 2 °C, receiving water and food “ad libitum”. The pre-clinical assay was approved by the CICUAL (Institutional Committee for the Care and Use of Laboratory Animals) of the School of Medicine, National University of La Plata, Argentina.

### 4.7. Pre-Clinical Assay: Skin Wound Model and Treatment in Mice

To evaluate the safety and efficiency of e-ucMSC therapy in wound healing, a modified mouse model with skin injury was used [29]. Surgical procedures were performed under anesthesia. Skin wounds were executed with a disposable dermatology biopsy punch. Incisions were made through the epidermis, dermis and subcutaneous tissue, leaving the deep fascia intact. It is necessary to take into account that mice are species characterized by the rapid healing of skin wounds; therefore, we needed to create a second injury over the first one, one week apart, so that wound healing by second intention allowed us to observe differences in the wound healing process between the groups. Animals were randomly distributed into 2 experimental groups (*n* = 6 in each group): group I (control), with skin injury injected only with 0.1 mL PBS, and group II, with skin injury and treatment with xenogeneic transplantation by injection of e-ucMSCs (dose of 250,000 cells/0.1 mL PBS) into the subcutaneous cellular tissue at the edges of the injury. 

In addition, we know that mouse ucMSCs express the Qa2 molecule, a functional and structural homolog of the human HLA-G protein [25]. By a previously published assay, we demonstrated the tolerance of human ucMSCs in mice by in vivo xenotransplantation of the cells [11].

Wound evolution: Daily observation of the mouse wound was performed by 2 independent observers until the healing process was complete and the hair had fully recovered. Wound healing was photographed on different days for subsequent analysis to measure the wound healing area and calculate the percentage of wound healing rate using Fiji Software (https://fiji.sc/ accessed on 20 December 2023).

Evaluation of graft rejection: To determine the immune response of mice to re-exposure to xenogeneic e-ucMSCs, a multiple-dose assay was performed. For this, a group of mice with skin injuries was injected 4 times, each with 250,000 cells/0.1 mL PBS, at 0, 2, 7 and 15 days. The wound was observed daily by two independent monitors to evaluate the local reaction. Forty-eight hours after the last dose, skin samples were collected for histological analysis of lymphocyte infiltration.

### 4.8. Clinical Trial 

Severe and deep leg injuries of horses were treated with e-ucMSCs. Treatment of the injured equines was performed with the approval of the veterinary staff in charge of the equines at the Horse Ranch. Between 7 and 10 days after the injury occurred, and prior to injection with the cells, the lesions were treated with the routine treatment for this type of complex wound, which includes cleaning and disinfection of the affected area, antibiotics, anti-inflammatory drugs, anti-myiasis and anti-fungal agents. It is important to note that the area must not show signs of infection at the time of cell injection. In this first phase of the clinical trial, four horses of both sexes were selected based on the severity of their lesions. The horses selected were those with large wounds, over 10 cm in surface area, deep enough to affect several types of tissues, such as the epidermis, dermis, connective tissue, joints, tendons and muscle. Another important aspect that was considered was that inadequate healing of these deep wounds could lead to the formation of scar tissue with the loss of tissue function. This could cause secondary or disabling complications in these horses, which could be avoided with the cell treatment we propose to use in this work.

To evaluate the safety and efficiency of equine cell therapy in wound healing, deep wounds on the legs were treated with allogeneic transplantation of e-ucMSCs (dose of 5 or 10 × 10^6^ e-ucMSCs in PBS). The cells were injected at the edges of the lesions in the epidermal and dermal tissues, as well as into the involved muscles and tendons. Depending on the extent and severity of the injury, a single dose was applied, or in other cases, we injected 2 doses of cells (separated by an interval of 15 days each) as the severity of the lesion required it. In all cases, blood tests were performed before the first injection and after each treatment to check all health parameters, including white blood cells. Wound healing was photographed every week for subsequent analysis. Finally, when the lesions were closed, ultrasound scans were performed to check the regeneration of the deeply involved tissues. All the injected cases were followed up with weekly observations until complete wound closure for up to 120 days.

### 4.9. Statistical Analysis

For immunomodulation assay values as well as for healing and hair recovery values in mice, the mean ± standard deviation (SD) of each group was calculated. The residuals fitted normal distribution and homogeneity of variance (Shapiro–Wilk test and F-test). Comparisons between groups were assessed using a T-test for unpaired data. Differences with *p* < 0.05 were considered significant. Statistical analyses were performed using GraphPad Prism 8 Software.

## Figures and Tables

**Figure 1 ijms-25-02350-f001:**
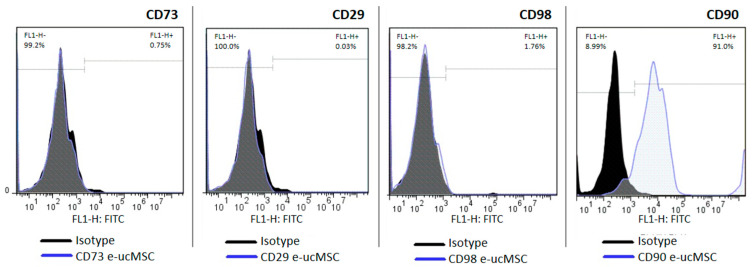
Characterization of e-ucMSCs. Flow cytometry analysis was performed. Cells were positive for CD90 and negative for CD73, CD29 and CD98.

**Figure 2 ijms-25-02350-f002:**
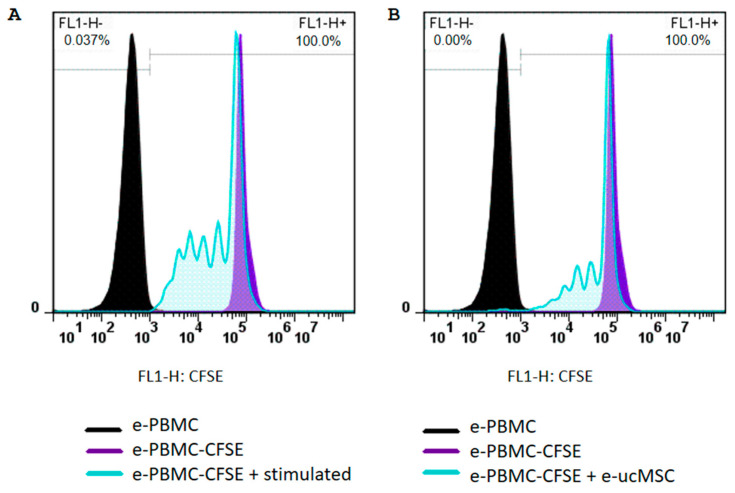
Inhibitory effect of e-ucMSCs on e-PBMC proliferation. (**A**) Normal proliferation cycles of activated e-PBMCs and (**B**) inhibition of proliferation cycles of e-PBMCs with co-culture with e-ucMSCs.

**Figure 3 ijms-25-02350-f003:**
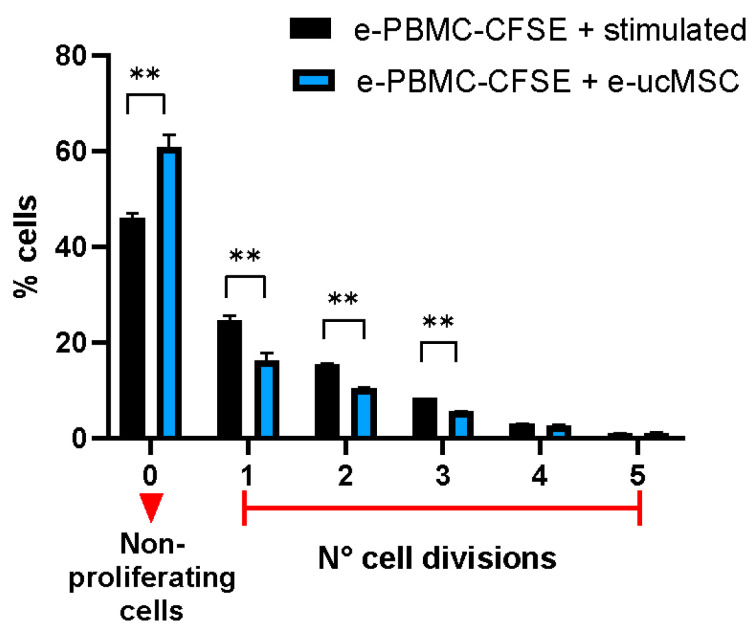
Effect of e-ucMSCs on the percentage of e-PBMCs in each cell division. ** (*p* < 0.01).

**Figure 4 ijms-25-02350-f004:**
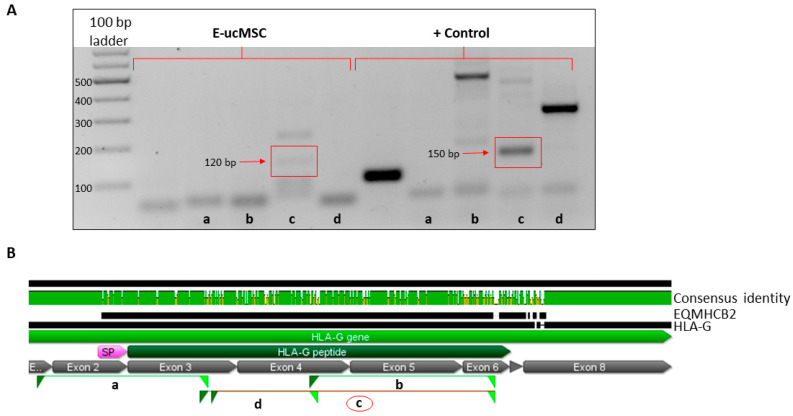
Detection of a similar HLA-G protein in e-ucMSCs. (**A**) Agarose gel revealing RT-PCR results. The expression of a similar HLAG mRNA with 120pb (red rectangle) in e-ucMSCs compared with the positive control (JEG-3). (**B**) Sequence alignment of the horse gene (*EQMHCB2*) with the human one (*HLA-G*). The different primer pairs are indicated as a, b, c and d. The c (red circle) indicates which exons of the gene it hits.

**Figure 5 ijms-25-02350-f005:**
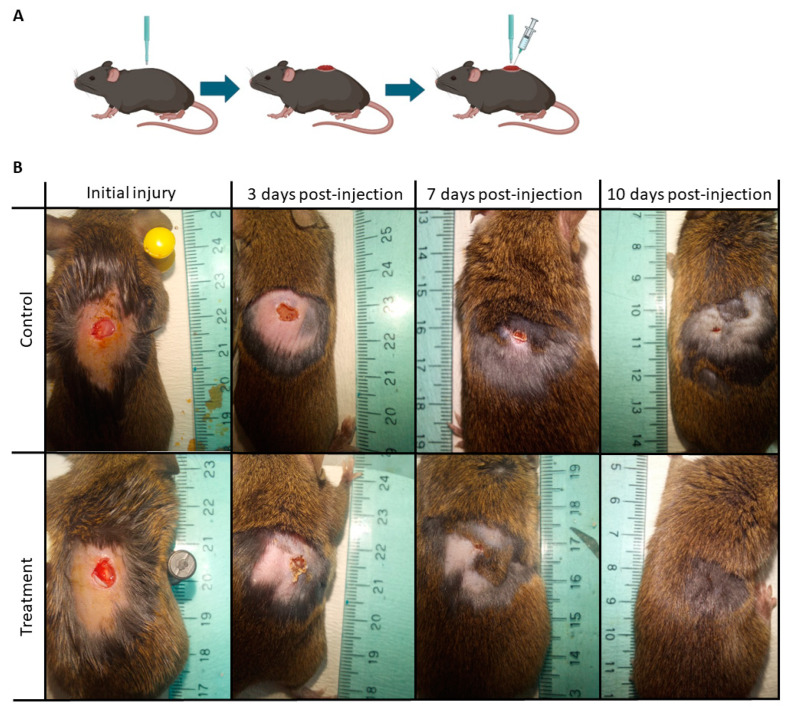
Xenograft transplantation: pre-clinical assay in mice. (**A**) Diagram of the mouse skin wound model by second intention. (**B**) Wound healing process. The images on the left show the initial injury and then the progress of wound healing 3, 7 and 10 days after injection. Control: injection only with PBS; treatment: injection with 250,000 e-ucMSCs/100 µL of PBS.

**Figure 6 ijms-25-02350-f006:**
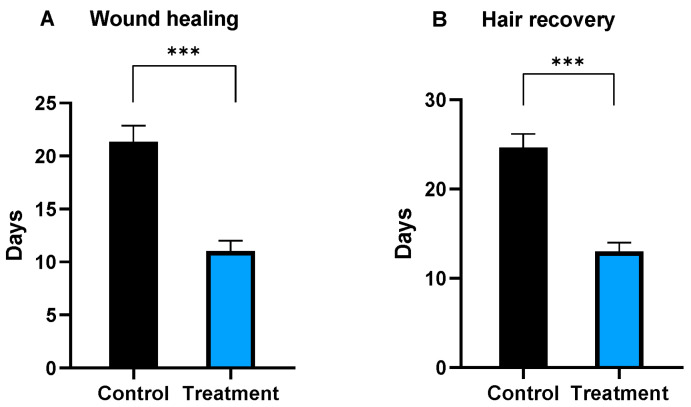
Effect of xenograft and allogeneic transplantation in mice. (**A**) Days for complete wound healing. (**B**) Days for full hair recovery. Control: mice injected with PBS, treatment: mice injected with one dose of e-ucMSCs. *** (*p* < 0.001).

**Figure 7 ijms-25-02350-f007:**
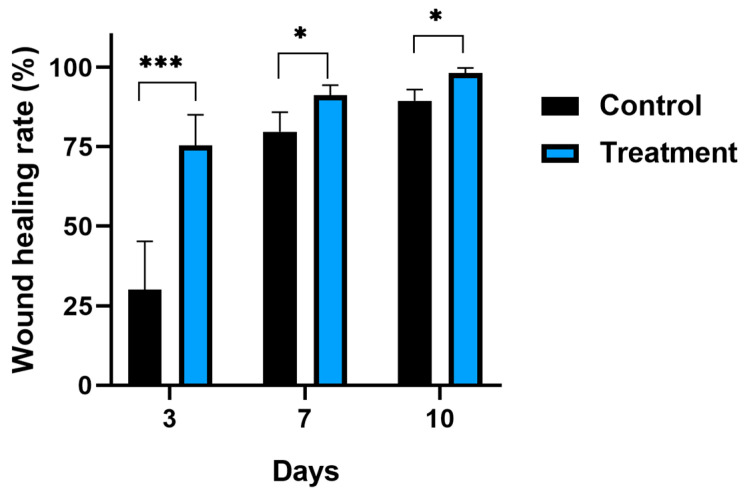
Percentage of wound healing rate at different time points. Control: mice injected with PBS. Treatment: mice injected with one dose of e-ucMSCs. * (*p* < 0.05) and *** (*p* < 0.001).

**Figure 8 ijms-25-02350-f008:**
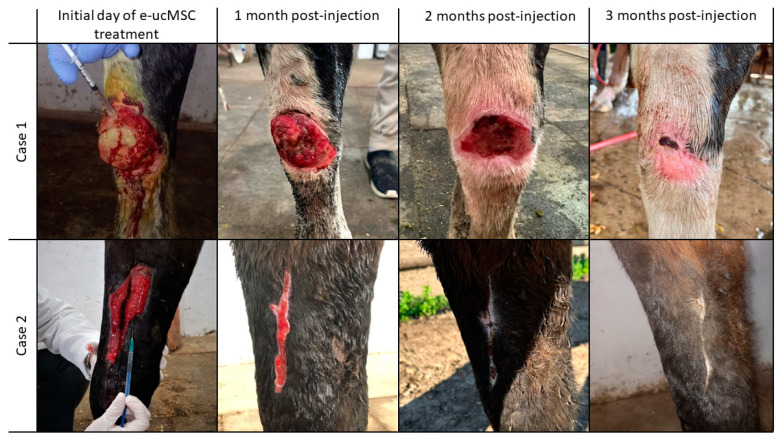
Clinical assay in injured horses. Two of the most severe cases with different anatomo-pathologies are presented to demonstrate the treatment with e-ucMSCs. The lesions involve the skin, connective tissue and muscles. The images on the right show the initial day of treatment, and those on the left show the progress of wound closure over 3 months.

**Table 1 ijms-25-02350-t001:** Primer sets used to search for homologs of any of the different HLA-G isoforms in e-ucMSCs.

Oligo Set	Oligo Name	5′ - Oligo Seq - 3′
a	Ex0 fw	CCTGGACTCACACGGAAACT
	257 rv	TGTTCCGTGTCTCCTCTTCC
b	G5U522 fw	CAATGTGGCTGAACAAAGGAGAG
	963 rv	GCAGCTCCAGTGACTACAGC
c	274c fw	CCACGCACAGACTGACAG
	963 rv	GCAGCTCCAGTGACTACAGC
d	257 fw	GGAAGAGGAGACACGGAACA
	526 rv	CCTTTGTTCAGCCACATTGG

**Table 2 ijms-25-02350-t002:** Representative blood analysis of case 2. Blood count and WBC relative percentage are shown, compared at different time points. RBCs: red blood cells. WBCs: white blood cells.

	Case 2	Before e-ucMSC Treatment	15- Days after First Dose	15- Days after Second Dose
Blood Count	RBC	10.2 × 10^6^/mm^3^	9.8 × 10^6^/mm^3^	9.9 × 10^6^/mm^3^
	WBC	11.3 × 10^3^/mm^3^	9 × 10^3^/mm^3^	8.1 × 10^3^/mm^3^
	Haemoglobin	16.3 g/dL	16.6 g/dL	16.8 g/dL
WBC Relative Percentage (%)	Segmented neutrophils	55	53	55
	Eosinophils	1	2	1
	Basophils	0	0	0
	Lymphocytes	43	43	43
	Monocytes	1	2	1

## Data Availability

Data is contained within the article and Appendix A.

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
