# Peer review of "Therapeutic Potential in Wound Healing of Allogeneic Use of Equine Umbilical Cord Mesenchymal Stem Cells"

_ijms, 2024, doi:10.3390/ijms25042350_

Round 1

Reviewer 1 Report

Comments and Suggestions for Authors

This is a rather interesting and important topic. Although there are many publications into wound healing in horses, the information on the use of equine mesenchymal cells in allogenic wound healing is (to my knowledge) scarce. The manuscript has an acceptable form, although I encourage the Authors to re-check it, possibly by a native or native-like speaker (there are some errors like 'characterization' instead of 'characteristics'). There are some features which need addressing by the Authors. Please, find them below.

- The visual side of this manuscript is unacceptable at its current state. Figures need to be re-uploaded in a more optimal resolution and form, since they look like they have been cropped out of MS Word. Some figures (like Figure 4) need re-arrangement, since the text is barely readable or hidden under some chunks of picture(s).
- Figure 7: it seems unclear to me why the t-test was performed instead of repeated measures ANOVA (RM-ANOVA). Utilizing the latter would be a more optimal practice, since the test would compare the difference between the two trends over the specified timepoints (time + time*group) instead of pairwise comparison of the points between the groups. Descriptive statistics for this analysis are: estimated marginal means and SE (or 95% CI of the means). Moreover, could the Authors explain why the hair recovery was skipped in this analysis? It could be seen in the previous figure.
- As the Authors chose the t-test for comparisons, I would recommend to show standard deviation instead of standard error. Standard error is associated with the estimation of the means per se, while standard deviation addresses the dispersion of the results in each group.

-Have the assumptions of the statistical tests been met? Was it checked? The distributions in both groups should resemble normal distribution. This could be checked with the Shapiro-Wilk test (or, with some experience, based on histograms and skewness/kurtosis). More importantly, the variance in both groups should be similar. The Levene test remains one of the most powerful tests to check for homoscedasticity in case of this dataset.

- I would swap bar plots for box plots for clearer visualization of the dispersion.
- It is written that the horses were selected based on the severity of their injuries. Could this topic be expanded for the Readers? The criteria should be transparent.

I vote for granting the Authors the rights to provide a major revision of this manuscript.

Comments on the Quality of English Language

The manuscript should be checked by a native or native-like English speaker since some words are used inappropriately in the text.

Reviewer 2 Report

Comments and Suggestions for Authors

EVALUATION

In the manuscript titled "Therapeutic potential in wound healing of allogeneic use of equine umbilical cord mesenchymal stem cells.", the authors used E-ucMSC as potential use to accelerate healing of equine legs wounds. Several studies have been done including immunomodulation culture assay, preclinical assays and many more. The author stated that the ucMSC could be allogeneically used for wound healing. However, we believe that the quality of the manuscript would be greatly improved if the authors could provide more in-depth information in an easier way for readability. Moreover, several comments below need to be addressed before published.

Major comments.

1.      Regarding the wound healing study, Figure 5 shows the comparison between control-group and treated-group. However, the authors only showed the images post-injection. We recommend adding the initial wound as the starting point for exact comparison. Moreover, normal wounds without infection will heal faster due to the body self-healing mechanism (we can see from the control group at day10 post-injection). Could the authors explain or show that the healing of treated-group is solely due to treatment with e-ucMSC. Also, rodents have loose skin and due to the elasticity of the back skin, the tension around wound is mostly reduced and trigger wound contraction, hence healed. Therefore, it is quite difficult to distinguish whether the healing is due to the internal factor or due to the treatment.

2.      We recommend adding ethical clearance for current studies due to the in vivo study (the use of animal for research).

Minor comments.

1.      Some typos can be found in manuscripts. Please do recheck thoroughly.

2.      We recommend rechecking some grammatical errors.

Round 2

Reviewer 1 Report

Comments and Suggestions for Authors

I would like to thank the Authors for revising the manuscript. I realize that, as noted, it is too late to revise the design of the study. The quality of featured images has drastically improved since the last version. Moreover, I believe that the study will gather citations as it may be a food for thought when designing more complex experiments into the matter of this topic. As a last remark, I would advise the Authors to indicate that the study should be deemed as preliminary, with commentary on its drawbacks in the discussion - as an icing on the cake before proceeding the manuscript further.

Author Response

Dear Reviewer,
We appreciate your work to improve this article. We are pleased to see that you found the corrections satisfactory. Taking into account your last remark we have added this last idea in the discussion between the end of page 10 and the beginning of page 11.
We believe that all your comments were of great help in shaping this final work.
Thank you very much.
Yours sincerely,

Reviewer 2 Report

Comments and Suggestions for Authors

good to go

Author Response

Dear Reviewer,

Thank you very much.

Yours sincerely.